# Study of Transcriptomic Analysis of Yak (*Bos grunniens*) and Cattle (*Bos taurus*) Pulmonary Artery Smooth Muscle Cells under Oxygen Concentration Gradients and Differences in Their Lung Histology and Expression of Pyruvate Dehydrogenase Kinase 1-Related Factors

**DOI:** 10.3390/ani13223450

**Published:** 2023-11-08

**Authors:** Yiyang Zhang, Manlin Zhou, Yuxin Liang, Rui Li, Lan Zhang, Shuwu Chen, Kun Yang, Haie Ding, Xiao Tan, Qian Zhang, Zilin Qiao

**Affiliations:** 1Engineering Research Center of Key Technology and Industrialization of Cell-Based Vaccine, Ministry of Education, Northwest Minzu University, Lanzhou 730030, China; zyy15890182033@163.com (Y.Z.); 15888533443@163.com (M.Z.); yunzhai4419@163.com (R.L.); 15095303494@163.com (S.C.); qzl13639315431@163.com (Z.Q.); 2Gansu Tech Innovation Center of Animal Cell, Biomedical Research Center, Northwest Minzu University, Lanzhou 730030, China; 3Key Laboratory of Biotechnology and Bioengineering of State Ethnic Affairs Commission, Biomedical Research Center, Northwest Minzu University, Lanzhou 730030, China; 4Life Science and Engineering College, Northwest Minzu University, Lanzhou 730030, China; 18405877207@163.com (Y.L.); 18693259358@163.com (L.Z.); wyyx20230825@126.com (H.D.); wyyx1157@126.com (X.T.); 5College of Veterinary Medicine, Gansu Agricultural University, Lanzhou 730070, China; zq880204@126.com

**Keywords:** yak, cattle, PASMCs, different oxygen concentrations, transcriptomic analysis, lung histology, hypoxic differential factor, expression distribution

## Abstract

**Simple Summary:**

The lung is a key organ that exhibits adaptive changes in response to high altitude in mammals. The prolonged presence of lowland cattle at high altitude leads to the abnormal proliferation of pulmonary vascular smooth muscle cells, resulting in pulmonary vascular remodeling, whereas the underlying molecular mechanisms in yaks, a representative model for mammalian high-altitude acclimatization studies, remain unknown. In this manuscript, we investigated the transcriptomic analysis of yak and cattle pulmonary artery smooth muscle cells in vitro at different oxygen concentrations (1%, 10%, and 20%) and in vivo to observe the lung tissue structure as well as the distribution of PDK1, HIF-1α, and VEGF and differences in their expression in the lungs of plateau yaks and plains cattle. The results showed that the HIF-1 signaling pathway, glucose metabolism pathway, and related factors (HK2/PGK1/ALDOA/ALDH1A3/EHHADH) were closely related to each other, that there were obvious differences between yak lung tissues and those of plains cattle, and that there might be a regulatory relationship between the differences in the distribution and expression of PDK1, HIF-1α, and VEGF and the adaptation of yak lungs to the plateau hypoxic environment. The differences in the distribution and expression of PDK1, HIF-1α, and VEGF may be related to the adaptation of yak lungs to the plateau hypoxic environment, which provides basic information for studying the mechanism of hypoxia adaptation in yaks.

**Abstract:**

The aim of this study was to investigate the molecular mechanisms by which hypoxia affects the biological behavior of yak PASMCs, the changes in the histological structure of yak and cattle lungs, and the relationships and regulatory roles that exist regarding the differences in the distribution and expression of PDK1 and its hypoxia-associated factors screened for their role in the adaptation of yak lungs to the plateau hypoxic environment. The results showed that, at the level of transcriptome sequencing, the molecular regulatory mechanisms of the HIF-1 signaling pathway, glucose metabolism pathway, and related factors (HK2/PGK1/ENO1/ENO3/ALDOC/ALDOA) may be closely related to the adaptation of yaks to the hypoxic environment of the plateau; at the tissue level, the presence of filled alveoli and semi-filled alveoli, thicker alveolar septa and basement membranes, a large number of erythrocytes, capillary distribution, and collagen fibers accounted for all levels of fine bronchioles in the lungs of yaks as compared to cattle. A higher percentage of goblet cells was found in the fine bronchioles of yaks, and PDK1, HIF-1α, and VEGF were predominantly distributed and expressed in the monolayers of ciliated columnar epithelium in the branches of the terminal fine bronchioles of yak and cattle lungs, with a small amount of it distributed in the alveolar septa; at the molecular level, the differences in PDK1 mRNA relative expression in the lungs of adult yaks and cattle were not significant (*p* > 0.05), the differences in HIF-1α and VEGF mRNA relative expression were significant (*p* < 0.05), and the expression of PDK1 and HIF-1α proteins in adult yaks was stronger than that in adult cattle. PDK1 and HIF-1α proteins were more strongly expressed in adult yaks than in adult cattle, and the difference was highly significant (*p* < 0.01); the relative expression of VEGF proteins was not significantly different between adult yaks and cattle (*p* > 0.05). The possible regulatory relationship between the above results and the adaptation of yak lungs to the plateau hypoxic environment paves the way for the regulatory mechanisms of PDK1, HIF-1α, and VEGF, and provides basic information for studying the mechanism of hypoxic adaptation of yaks in the plateau. At the same time, it provides a reference for human hypoxia adaptation and a target for the prevention and treatment of plateau diseases in humans and plateau animals.

## 1. Introduction

The plateau region has special environmental characteristics, such as strong ultraviolet radiation, a short grass-growing season, low temperature, and low oxygen. The yak (*Bos grunniens*) is a unique species on the Qinghai-Tibet Plateau with a strong lung function, strong foraging ability, and high metabolic ability, a series of characteristics that are adapted to the plateau region [1,2,3]. In addition, yaks are able to pass on their stable adaptations to the next generation, making it an ideal species to study hypoxic adaptation in the plateau, where the underlying molecular mechanisms are still largely unknown [4,5,6]. Therefore, research on the mechanism of hypoxic adaptation in plateau animals is an important scientific problem that needs to be solved urgently.

The lung is the main organ for gas exchange between the respiratory system and the cardiovascular system, and the partial pressure of oxygen required for survival is 4–18%. Studies have shown that the sensitivity of animal lung tissue to hypoxia is closely related to the smooth muscle content of the pulmonary vascular wall, and the higher the smooth muscle content, the higher the sensitivity to hypoxia [7]. Under normal physiological conditions, the structure and functional status of pulmonary circulation are affected by hypoxia, which will lead to changes in pulmonary vascular diameter, pulmonary vascular remodeling, and hypoxic pulmonary hypertension (PAH) [8,9]. Furthermore, abnormal proliferation of PASMCs is a major cause of pulmonary vascular remodeling [10,11,12]. Therefore, there is a great need to study the state of PASMCs under hypoxia. Under normal physiological conditions, these cells maintain low proliferation, low migration, and high contractility and express a specific set of cytoskeletal and contractile proteins. In the dedifferentiated state, PASMCs have low contractility and high proliferation, migration, and secretion of extracellular matrix (ECM). PASMCs are also involved in apoptosis, antioxidant activity, and vascular composition [13], whereas hyperproliferation and antiapoptosis of PASMCs promote pathophysiologic changes in PAH [14]. Therefore, an in-depth study of the molecular mechanisms of PASMC dedifferentiation is important for the understanding and prevention of cardiovascular diseases.

Prolonged exposure of animals to highland hypoxia triggers a variety of pathological mechanisms, and hypoxia affects organisms by activating multiple mechanisms at the physiological, cellular, and molecular levels. One of the diseases triggered is hypoxic pulmonary hypertension (HPH). Some studies have shown that lowland mammals are highly susceptible to severe pulmonary hypertension when they live or are kept for long periods in the hypoxic regions of the plateau [15]. However, highland yaks are protected from severe hypoxia-induced PAH, and the morphological structure of their respiratory and cardiovascular tissues and organs has evolved to adapt to the survival characteristics of high-altitude hypoxic environments [16,17,18]. Research has shown that the adaptability of yaks to this harsh environment is closely related to their lungs [16,19]. Yaks have developed strong pulmonary vasculature, which increases the rate of oxygen exchange in the pulmonary artery vasculature and helps relieve pulmonary artery pressure [20]. The structural, organizational, and morphological differences between the pulmonary arteries of yaks and those of cattle have been observed, and it was found that the alveolar area per unit area of yaks is larger and the alveolar intervals are thinner. In addition, the thickness of the muscular layer of the membranes in the microarterioles of yaks and the percentage of the outer diameter of blood vascular tissue in yaks is smaller than those of cattle, and the thickness of the air-blood barrier in yaks is thinner. These structural features not only facilitate gas exchange, but also prevent the occurrence of pulmonary hypertension [21,22,23]. The physiological and anatomical characteristics of yaks allow them to extract and utilize oxygen more efficiently, ensuring their survival in hypoxic conditions. At the same time, their underlying molecular regulatory mechanisms play an important role.

Therefore, it is necessary to study the adaptation mechanisms of the pulmonary artery smooth muscle cells in plateau yaks and plains cattle under different oxygen concentrations and the differences that exist between their lung tissues. In this experiment, yak and cattle pulmonary artery smooth muscle cells under different oxygen concentration gradients were cultured in vitro. Thus, 1% O_2_ was selected for acute hypoxic stress experiments, and PASMCs treated with different oxygen concentrations were analyzed and compared by bioinformatics. At the same time, there is limited research on the bronchioles of yaks and cattle at all levels, and there are currently no studies of the regulatory relationship between PDK1 and the plateau adaptation mechanism of yaks screened by bioinformatics. Therefore, this study compared the structural differences between yak and cattle lung tissues and the distribution and expression of PDK1 and its related factors in adult yak and cattle lung tissues through H&E staining, Masson staining, PAS staining, immunohistochemistry staining, optical density analysis, etc. We speculated that there is a correlation between the functions of PDK1, HIF-1α, and VEGF and the adaptation of yaks to low-oxygen environments at high altitude. By detecting the mRNA and protein expression of PDK1 and its related factors through RT-qPCR and Western blotting, we aimed to further explore PDK1 and its related factors’ hypoxic regulatory relationship, and the mechanism of action of yaks’ high-altitude adaptations, in order to obtain a biological model of convergent evolution or adaptation to a high-altitude hypoxic environment between plateau native animals and plains animals, providing lessons for human hypoxia adaptation and targets for combating plateau diseases in humans and plateau animals.

## 2. Materials and Methods

### 2.1. Animal Ethics

All animals were experimented on under license number (DKY-B20151608) by the Institutional Animal Ethics and Use Committee of the College of Life Sciences and Engineering, Northwestern University for Nationalities. All procedures in this study were carried out in accordance with the Guidelines for the Care and Use of Laboratory Animals formulated by the Ministry of Science and Technology, China.

### 2.2. Sample Collection

The lungs of three healthy adult cattle were collected from the slaughterhouse in Tianshan Town, Chifeng City, Chifeng City, Inner Mongolia Autonomous Region (about 1000 m above sea level), and the lungs of three healthy adult yaks were collected from the slaughterhouse in Cooperative City, Lanzhou, China (about 3500 m above sea level). All of the yaks were about 3 years old, and the lungs were carefully clipped to about 2 cm^3^ to prevent extrusion and fixed in 4% paraformaldehyde. During sampling, the lungs were carefully cut into 2 cm^3^ blocks and fixed in 4% paraformaldehyde; 1 cm^3^ blocks were cut into tinfoil, transported in an ice box, and, finally, stored in a refrigerator at −80 °C. In a hypoxia incubator (CCL-050T-8, Esco Lifesciences Group, Shanghai, China), previously isolated and purified primary PASMCs from yaks and cattle were cultured in the laboratory at 1%, 10%, and 20% oxygen concentration for 48 h, respectively, and then harvested for subsequent experiments. Each group had three replicates.

### 2.3. RNA Extraction

Total RNA was extracted from PASMCs and adult yak and cattle lung tissues using Trizol (Ambion, Thermo Fisher Scientific—CN, Shanghai, China). NanoDrop ND-2000 spectrophotometer (Nano Drop, Wilmington, DE, USA) and Bioanalyzer 2100 (Agilent Technologies, Santa Clara, CA, USA) were used to monitor RNA concentration and integrity, respectively.

### 2.4. Library Construction and Sequencing

Small RNAs of 14–40 nt in length were first purified by polyacrylamide gel electrophoresis (PAGE), and then, specific adaptors were ligated to the purified small RNAs. The ligated RNA was reverse transcribed into a cDNA library. Finally, Illumina sequencing was performed on a genome analyzer for each library. The flowchart of library construction and sequencing is shown below (Figure 1).

### 2.5. Identification of Differentially Expressed Genes

In order to reveal the transcriptome differences between yaks and cattle under different oxygen-concentration conditions, differential expression analyses were performed using DESeq2 package software (1.20.0) (https://bioconductor.org/packages/release/bioc/html/DESeq2.html, accessed on 12 March 2022) with biological replicates. The standard conditions for differential gene screening were |log2(FoldChange)| ≥ 1&padj ≤ 0.05. DESeq2 provides statistical routines to determine differential expression in numerical gene expression data using a model based on negative binomial distribution. The resulting *p*-values were adjusted using the method of Benjamini and Hochberg to control for false discovery rates. Genes with adjusted *p*-values less than 0.05 found by DESeq2 were classified as differentially expressed genes. For edgeR (3.18.1) (https://bioconductor.org/packages/release/bioc/html/edgeR.html, accessed on 14 March 2022) without biological replicates, prior to differential expression analysis, the edgeR package adjusted the number of reads in each sequenced library by using a scaling normalization factor. Differential expression analyses were performed for both conditions using the edgeR package (3.18.1) (https://bioconductor.org/packages/release/bioc/html/edgeR.html, accessed on 14 March 2022). *p*-values were adjusted using the Benjamini and Hochberg method. The corrected *p*-value was 0.05, and an absolute fold change of 2 was used as the threshold for significant differential expression.

### 2.6. Transcriptome Sequencing Data Processing and Bioinformatics Analysis

#### 2.6.1. GO Enrichment Analysis of Differentially Expressed Genes

Gene ontology (GO) enrichment analysis of differentially expressed genes was implemented by the clusterProfiler R package (3.8.1) (https://bioconductor.org/packages/release/bioc/html/clusterProfiler.html, accessed on 18 March 2022), in which gene length bias was corrected. GO terms with corrected *p*-value less than 0.05 were considered significantly enriched by differential expressed genes.

#### 2.6.2. KEGG Enrichment Analysis of Differentially Expressed Genes

KEGG is a database resource for understanding high-level functions and utilities of the biological system, such as the cell, the organism, and the ecosystem, from molecular-level information, especially large-scale molecular datasets generated by genome sequencing and other high-throughput experimental technologies. We used clusterProfiler R package (3.8.1) (https://bioconductor.org/packages/release/bioc/html/clusterProfiler.html, accessed on 18 March 2022) to test the statistical enrichment of differential expression genes in KEGG pathways.

### 2.7. Lung Tissue Sample Processing, Staining, and Testing

#### 2.7.1. Preparation of Lung Tissue Sections

Yak lung tissues of different ages were removed from the fixative, trimmed into 0.5 cm^3^ pieces with smooth edges, and placed into a fixative box, rinsed under running water for 24 h, dehydrated, embedded in paraffin wax, and sliced to form paraffin sections with a thickness of 5 µm, which were then spread out, fished out, and preserved after baking, and then used for the subsequent staining of the tissues.

#### 2.7.2. H&E Staining

Tissue sections were baked on a 45 °C baking machine for 2 h. Then, the sections were dewaxed and hydrated by xylene and gradient alcohol (10 min each), washed with running water for 5 min, nuclear stained with hematoxylin for 6 min, washed with running water for 15 min, differentiated with 1% hydrochloric acid alcohol solution for 3 s, and then washed with running water until the sections became blue. Then, the process was halted. Subsequently, the sections were stained with eosin staining solution for 8 min, dehydrated with gradient alcohol (5 min each), rendered transparent with xylene, and then gelled with neutral gum. After staining, the sections were dehydrated with gradient alcohol (5 min each), rendered transparent with xylene, then sealed with neutral gum, and stored after natural drying.

#### 2.7.3. Masson Staining

The sections were routinely dewaxed to water followed by ferruginous staining for 5–10 min. Then, the sections were placed in acidic ethanol differentiation solution for 5–15 s, subjected to Masson bluing solution flooding for 3–5 min, and then placed in Lichun red compound solution for 5–10 min. At the end of each step, we washed the sections with distilled water for 1 min, and then washed them with a weak acidic working solution for 1 min, with phosphomolybdic acid staining solution for 1 min, and with a weak acidic working solution for 1 min. Then, we washed the sections with aniline blue dye solution for 1–2 min and with weak acid working solution for 1 min. Then, 95% and anhydrous ethanol rapid dehydration was performed, and the sections were rendered transparent with xylene three times, 2 min/time, using sealing film.

#### 2.7.4. PAS Staining

Sections were routinely dewaxed to water, washed in tap water for 2–3 min, and washed twice with distilled water. Then, they were washed using oxidizing agents for 5–8 min, washed in tap water for 2–3 min, and washed twice with distilled water. Then, the sections were immersed in Schiff staining solution protected from light for 10–20 min and washed in tap water for 10 min. Hematoxylin nucleo staining was carried out for 1–2 min; then, the sections were placed in acidic differentiation solution for 2–5 s and counterblue in tap water for 10–15 min. Dehydration and transparency were performed routinely using neutral gum to seal film.

#### 2.7.5. Immunohistochemistry

Immunohistochemical staining was performed based on the SP assay kit. Tissue sections were deparaffinized in xylene, rehydrated with gradient ethanol, rinsed in PBS, and heated in 0.01 mol/L sodium citrate buffer (pH = 6.0) to repair the antigens (15 min in the microwave oven). Endogenous peroxidase was inactivated using 3% hydrogen peroxide at 37 °C and added dropwise to the sections, which were incubated for 10 min at 37 °C in a warm box, then dumped. Primary antibodies (1:500 PDK1, 1:500HIF-1α, 1:400 VEGF dilution, blank control replaced by PBS) were added dropwise, and sections were placed into a wet box, and then incubated at 4 °C overnight for staining of DAB substrate and re-staining with hematoxylin. Between the two steps, the sections were washed with PBS, dehydrated, rendered transparent, and sealed with neutral gum.

### 2.8. RT-qPCR Assay

#### 2.8.1. Synthesis of Gene Primers

According to the sequences of HK2, PGK1, ALDOA, ALDH1A3, EHHADH, PDK1, HIF-1α, and VEGF genes in GenBank, Primer-BLAST was used to design the primers for HK2, PGK1, ALDOA, ALDH1A3, EHHADH, PDK1, HIF-1α, and VEGF genes, and, at the same time, β-actin (actin beta, ACTB) gene was used as the internal reference gene. The information regarding the primer sequences is shown in Table 1. The data were sent to Hunan Accurate Bioengineering Co. (Hunan, China) for synthesis.

#### 2.8.2. Reverse Transcription

The above-mentioned yak and cattle PASMCs and lung tissue RNA were reverse transcribed with 1 μg of total RNA, and this c DNA was used as a template for qRT-PCR, using the following reaction system (20 μL): 8.2 μL of RNAse-free H_2_O, 0.8 μL each of upstream and downstream bows of 0.2 μmol/L, 2× Universal SYBR Green Fast qPCR Mix 10 μL, 1 μL of cDNA. The reaction conditions were as follows: pre-denaturation at 95 °C for 30 s, denaturation at 95 °C for 5 s, annealing at 60 °C for 35 s, and a total of 40 cycles. The qRT-PCR reaction was performed in a BIO-RAD CFX96 fluorescence quantitative PCR instrument, and the internal reference β-actin normalized the obtained values. qRT-PCR results were analyzed by 2^−∆∆Ct^ operation.

### 2.9. Western Blot Assay

#### 2.9.1. Protein Sample Preparation

A total of 0.1 g of lung tissue from each adult yak and cattle specimen was placed in a centrifuge tube. Ophthalmic scissors were used to fully shear the tissue. Then, 1000 μL of RIPA lysate (1% PMSF was added to the lysate) was added and the tissue was lysed on ice for 30 min and centrifuged at 4 °C and 12,000 rpm for 10 min. Then, the supernatant was extracted, the protein concentration was determined by the BCA method, and the protein concentration of each sample was adjusted to 1.5 μg/μL.

#### 2.9.2. Western Blot Analysis

After SDS-PAGE gel electrophoresis, the proteins were transferred to the PVDF membrane by the wet transfer method, and then the membrane was closed with TBST solution containing 5% skimmed milk for 2 h at room temperature; the primary antibodies (1:2000 PDK1; 1:1000 HIF-1α; 1:1000 VEGF) were added and incubated at 4 °C overnight, and the membrane was washed three times with TBST (five times each time).

### 2.10. Statistical Analysis

All measured data were expressed as mean ± standard error of measurement (SEM), and data were analyzed using GraphPad Prism 8 software for conformity to normal distribution and chi-squared. Then, comparative analyses were performed using Duncan’s one-way (ANOVA) method.

## 3. Results

### 3.1. Gene Expression Distribution

Due to the effect of sequencing depth and gene length, the gene expression values of RNA-Seq were expressed by fragments per kilobase of exon per million fragments mapped (FPKM), which was successively corrected for sequencing depth and gene length. In this experiment, the sequencing depth was 6G, with three replicates in each group. After firstly calculating the expression values of all the genes in each sample (FPKM), the distributions of the different gene expression levels in samples from the yaks and yellow cattle were demonstrated by box plots (Figure 2). It can be seen from the figure that the reproducibility between parallel samples was good.

### 3.2. Inter-Sample Correlation

The correlation of gene expression levels between samples is an important indicator of the reliability of the experiment and the appropriateness of the sample selection. The closer the correlation coefficient is to one, the greater the similarity of expression patterns between samples. The R^2^ between biologically replicated samples should be at least greater than 0.8. The higher the correlation coefficient between samples, the closer their expression patterns. The correlation coefficients of samples within and between groups were calculated based on the FPKM values of all genes in each sample and plotted as a heat map. This can allow one to observe the differences between samples from cattle and yaks and to observe good repetition of samples within groups (Figure 3).

### 3.3. Principal Component Analysis

Principal component analysis (PCA) is also commonly used to assess between-group differences and within-group sample duplication. PCA uses linear algebra calculations for dimensionality reduction and principal component extraction for tens of thousands of genetic variables. We performed PCA analysis on the gene expression values (FPKM) of all the samples, as shown in the following figure (Figure 4). In the PCA plot, the samples are scattered between groups and the samples within groups are clustered together, indicating that the repetition within groups is relatively good and the sample data are very similar, while there is good differentiation between groups. Hypoxia 1_2 had large within-sample group differences and was deleted and not included in subsequent experimental analyses.

### 3.4. Transcriptome Differential Gene Analysis

The results showed that 997 differential genes were obtained from cattle PASMCs in the hypoxia 1 vs. control comparison. Of these, 795 were upregulated and 202 were downregulated; 642 were obtained in the hypoxia 10 vs. hypoxia 1 comparison, of which 128 were upregulated and 514 were downregulated; and 365 were obtained in the hypoxia 10 vs. control comparison, of which 251 were upregulated and 114 were downregulated (Figure 5A). In addition, using a Wayne diagram to show the overlap of differential genes between different comparison combinations in cattle, we could filter out 20 comparison combinations of common or unique differential genes (Figure 5B). Yak PASMCs yielded 1080 differential genes in the hypoxia 1 vs. control comparisons, of which 743 were upregulated and 337 were downregulated; PASMCs yielded 1778 differential genes in the hypoxia 10 vs. hypoxia 1 comparisons, of which 797 were upregulated and 981 were downregulated; and PASMCs yielded 757 differential genes in the hypoxia 10 vs. control comparisons, of which 439 were upregulated and 318 were downregulated (Figure 5A). In addition, using a Wayne diagram to show the overlap of differentially expressed genes between different comparison combinations in the yak, it was possible to filter out 61 comparison combinations of shared or unique differentially expressed genes (Figure 5B). The clustering heat map of differentially expressed proteins (Figure 5C) showed that the samples of yak and cattle PASMCs at different oxygen concentrations had good biological reproducibility.

By combining the relevant literature, we screened the research pathways related to the direction of hypoxia adaptation research, and then randomly screened the DEGs for validation (Figure 6A,B). In correlation analysis, the closer R^2^ is to one, the better the correlation, the higher the agreement between RT-qPCR and RNA-Seq, and the more reliable the sequencing effect (Figure 6C–F).

### 3.5. Enrichment Analysis

#### 3.5.1. GO Function Enrichment Analysis

ClusterProfile software was used to perform GO function enrichment analysis on the differential gene set to analyze the up- and downregulated genes related to yak and cattle PASMCs with different oxygen concentrations. The differentially expressed up- and downregulated genes of hypoxia 1 vs. control, hypoxia 10 vs. control, and hypoxia 10 vs. hypoxia 1 oxygen concentrations were subjected to GO function analysis, and it was generally considered that *p* < 0.05 indicated significant enrichment. The results showed that DEGs of pulmonary artery smooth muscle cells from yaks with different oxygen concentrations and from cattle with different oxygen concentrations were enriched in three major categories, namely, molecular functions, biological processes, and cellular fractions, and the top 30 categories were selected to be displayed (Figure 7A–F). Of the cattle PASMCs with different oxygen concentrations (Figure 8A) in the BP functional category, the top 10 enriched GO categories were related to DAN metabolism, and the most enriched categories were the DNA metabolic process, DNA replication, and the monocarboxylic acid metabolic process. Among the CC functional categories, the most enriched categories were the extracellular region and nucleus. Among the MF functional categories, the most enriched categories were: hydrolase activity, acting on acid, nucleoside triphosphatase activity, pyrophosphatase activity, oxidative stress, pyrophosphatase activity, and oxidoreductase activity. Among yak PASMCs with different oxygen concentrations (Figure 8B), in the BP functional category, the top 10 enriched GO categories were related to DNA metabolism, of which the most enriched were the DNA metabolic process, G protein-coupled receptor signaling, and the nucleoside metabolic process. Among the CC functional categories, the most enriched were chromosomes and the extracellular region. Among the MF functional categories, the most enriched were: hydrolase activity, acting on acid, nucleoside triphosphatase activity, pyrophosphatase activity, pyrophosphatase activity, selenocyte activity, pyrophosphatase activity, and signaling receptor activity.

#### 3.5.2. KEGG Pathway Enrichment Analysis

To further understand the metabolic and signaling pathways in which these DEGs are involved, the KEGG public database was used for analysis in this study. KEGG analysis was performed on the differentially expressed genes in yaks and cattle at different oxygen concentrations (hypoxia 1 vs. control, hypoxia 10 vs. control, hypoxia 1 vs. hypoxia 10), and KEGG enrichment was significantly improved at Padi < 0.05. The most significant DEGs in hypoxia 1 vs. control, hypoxia 10 vs. control, and hypoxia 1 vs. hypoxia 10 were selected. The top 20 KEGG pathways of the DEGs were plotted in bar graphs to show this. The most enriched pathway DNA replication was among the top 20 most enriched pathways in the hypoxia 1 vs. control group (Figure 9A), the most enriched pathway cell cycle was among the top 20 most enriched pathways in the hypoxia 10 vs. control group (Figure 9B), and the most enriched steroid biosynthesis pathway among the top 20 most enriched pathways was in the hypoxia 1 vs. hypoxia 10 group (Figure 9C) for the cattle PASMCs. The most enriched pathway among the top 20 pathways in the hypoxia 1 vs. control and hypoxia 1 vs. hypoxia 10 groups for yak PASMCs was DNA replication (Figure 9D,F), and the most enriched pathway among the top 20 pathways in the hypoxia 10 vs. control group was steroid biosynthesis (Figure 9E).

### 3.6. Results of H&E Staining

H&E staining results showed that the lungs of adult cattle and yaks were complete in structure and organization, and the parenchyma of the lung tissue was obvious and divided into two parts, the conducting-region part and the respiratory part; the conducting-region part consisted of the small bronchioles, fine bronchioles, and terminal bronchioles, and the respiratory part consisted of respiratory bronchioles, alveolar ducts, alveolar sacs, and alveoli, and the development of fine bronchioles and alveolar parts of all levels was in a good condition. The various levels of pulmonary bronchioles in the cattle were sparse, while the peribronchiolar alveoli at all levels in the yaks were dense. The peribronchiolar alveoli at all levels in the adult cattle were predominantly filled alveoli, while the peribronchiolar alveoli at all levels in the adult yaks were both semi-filled and filled (Figure 10). At the same time, it was observed that the thickness of alveolar septa was thicker in yaks compared with that in cattle (Figure 10), and the percentage of septa in the peripheral alveoli of various levels of pulmonary bronchioles was higher in yaks than in yellow cattle, which was a highly significant difference (*p* < 0.01) as measured by Image-Pro Plus (Figure 11A). According to the arterial vascular classification criteria in Table 2 [24], the diameters of the accompanying pulmonary arterial vascular vessels of the pulmonary fine bronchioles at all levels in cattle and yaks were measured and classified, and in combination with the results of the measurements in Table 3, it was found that the type of accompanying pulmonary arterial vascular vessel of the pulmonary fine bronchioles at all levels of the two was pulmonary microneural arteries, and the average thickness of the intima-media of the accompanying pulmonary arterioles at all levels of the lungs of the cattle and the yaks was measured and compared with that of the cattle and the yaks using the ImageJ software (v1.51). It was found that the mean intima-media thickness of the accompanying pulmonary arteries was significantly greater in the yaks than in the cattle (*p* < 0.05) (Figure 11B).

### 3.7. Results of Masson Staining

The results of Masson staining showed that the results of peribronchiolar alveoli at all levels of fine bronchioles in adult cattle and yaks were consistent with the results of H&E, and it was also found that the alveoli around fine bronchioles at all levels and the accompanying pulmonary arteries were mostly semi-filled in yaks, and the alveolar septa thickness was thicker, while the alveoli far from fine bronchioles at all levels and pulmonary arteries were mostly filled (Figure 12); the percentage of alveolar septa in the alveoli surrounding fine bronchioles at all levels was found to be higher in yaks than in cattle (Figure 12), the percentage of alveolar septa in alveoli surrounding fine bronchioles at all levels was found to be higher in yaks than in cattle, and the difference was highly significant (*p* < 0.01) (Figure 13A). The results were consistent with those of H&E staining. The fine and terminal bronchi have the role of controlling air flow into the alveoli, mainly through diastole and smooth muscle contraction, while the respiratory fine bronchi are closely connected to the alveoli, which is an important oxygen exchange pathway. For this reason, the average thickness of smooth muscle at all levels of fine bronchioles was observed and measured, and it was found that the average thickness of smooth muscle at all levels of fine bronchioles in yaks was significantly greater than that in cattle (*p* < 0.01) (Figure 12 and Figure 13B); the percentage of collagen fibers in the fine bronchioles of all levels of the lungs of adult yaks was significantly higher than that of cattle (*p* < 0.01) (Figure 11C), suggesting that the high percentage of collagen fibers in the fine lung bronchioles at all levels in yaks may be related to better adaptation to the hypoxic environment.

### 3.8. Results of PAS Staining

The results of PAS staining showed that the PAS reaction of the goblet cells in the fine bronchial epithelium was positive in adult cattle and yaks (Figure 14), and it was also found that the percentage of goblet cells in the lung fine bronchioles of adult yaks was high compared with that of cattle. The difference was highly significant (*p* < 0.01) (Figure 15C), indicating that there was a large amount of glycogen in the goblet cells in the fine bronchial epithelium, which was beneficial for mucosal protection, whereas the goblet cells disappeared in the terminal fine bronchioles and respiratory fine bronchioles, and the PAS reaction was negative. It was observed that the PAS reaction was positive in the peribronchial smooth muscle at all levels of the lungs of adult cattle and yaks, and the positive expression in the peribronchial smooth muscle at all levels of the lungs was stronger in adult yaks than in cattle (Figure 14). This indicates that the content of myoglycogen in the peribronchial smooth muscle at all levels of the lung was higher in yaks than in cattle, and that the distribution of glycogen in the peribronchial smooth muscle at all levels of the lung was beneficial for the provision of energy for smooth muscle contraction and diastole in the respiratory period of yaks. The high glycogen content in the peribronchial smooth muscle at all levels of the fine bronchioles is beneficial in providing energy for smooth muscle contraction and diastole during respiration. The percentage of alveolar septa in the peribronchiolar alveoli at all levels of the lung was significantly higher in yaks than in cattle (*p* < 0.01) (Figure 15A), which was consistent with the results of H&E and Masson staining. As the basement membrane plays a supporting role for tissue structure, we measured the basement membrane thickness of pulmonary fine bronchioles and terminal fine bronchioles and found that the basement membrane thickness in adult yaks was significantly greater than that in cattle (*p* < 0.01). We speculate that the change in basement membrane thickness was related to the better adaptation of yaks to a hypoxic environment.

### 3.9. Results of Immunohistochemical Staining and Optical Density Analysis

According to the immunohistochemical results, PDK1, HIF-1α, and VEGF were mainly distributed in the monolayer columnar epithelium of the terminal fine bronchioles and their branches in the lungs of cattle and yaks, with a small amount of them distributed in the alveolar septa, and they were mainly expressed in the epithelial cells of the terminal fine bronchioles and their branches and in the alveolar septal cells. The positive expression of PDK1 was stronger in bovine lungs than in yak lungs, and the difference between the mean optical density values of PDK1 on terminal fine bronchioles and alveolar septa of yak and cattle lungs was highly significant (*p* < 0.01) (Figure 16A,E and Figure 17); according to the overall HIF-1α and VEGF positive expression results, adult yak lungs were stronger than cattle lungs, and the positive expression results of VEGF in yak terminal fine bronchioles were more strongly expressed. Further, the mean optical density values of HIF-1α and VEGF on terminal fine bronchioles and alveolar septa of yak and bovine lungs were significantly different (*p* < 0.05) (Figure 16B,C,F,G and Figure 17); as shown in Figure 16D,H, they were not expressed in the negative control adult yak and cattle lungs.

### 3.10. PDK1 and Its Related Factors’ mRNA Expression in Adult Cattle and Yak Lungs

From Figure 18, it can be seen that the PDK1 gene was expressed in both adult cattle and yak lungs, and there was no significant difference in the expression of PDK1 in adult cattle and yak lungs (*p* > 0.05) (Figure 18A); HIF-1α was expressed in both adult cattle and yak lungs, and in adult yak lungs, the expression of HIF-1α was higher than that in adult bovine lungs. The difference was significant (*p* < 0.05) (Figure 18B); the relative mRNA expression of VEGF differed significantly (*p* < 0.05) between adult bovine and yak lungs (Figure 18C), and the difference in expression was similar to that of HIF-1α.

### 3.11. Protein Expression of PDK1 and Related Factors in the Lungs of Adult Yaks and Cattle

From Figure 19, it can be seen that PDK1 and HIF-1α proteins were expressed in the lungs of adult cattle and yaks, and the expression in adult yaks was stronger than that in adult cattle. The difference was highly significant (*p* < 0.01); the relative protein expression of VEGF was not significantly different between adult yaks and cattle (*p* > 0.05).

## 4. Discussion

The low partial pressure of oxygen at high altitude has serious implications for the survival and development of humans and other animals. As a result of natural selection, animals native to the Tibetan Plateau show genetic adaptations to this extreme environment. Therefore, the yak is an ideal animal model to study hypoxia-related molecular ecology and pathology [25]. RNA-Seq technology has been widely used as a method to screen functional genes in domestic species due to its characteristics of low bias and high reproducibility. The sequencing of the yak genome was first published in 2012 [26], which laid an important foundation for subsequent yak-related studies. Among the previous studies on hypoxia−adapted transcription in yaks, most of the previous studies on the transcriptome of yaks (*Bos grunniens*) focused on reproduction and nutrition [27,28,29,30]. The studies on the lung transcriptome of high-altitude yaks and low−altitude cattle (*Bos taurus*) focused on exploring the potential mechanisms of yak adaptation to high-altitude environments [25,26,31,32], but no studies have ever been conducted at the cellular level to reveal the mechanism of yaks’ adaptation to hypoxia. Therefore, by subjecting PASMCs from high-altitude yaks and low-altitude cattle to acute stress treatment with different oxygen concentration gradients (1%, 10%, 20%) and thus performing RNA-Seq transcriptomic analysis, we can comprehensively analyze the hypoxic adaptation of yak PASMCs at the cellular level and thereby explore the potential mechanisms of yak adaptation to high-altitude environments.

Guan et al. [33] found differences in microRNA transcriptome levels between yaks and cattle and showed that differentially expressed genes existed in yak and plains yak tissues sequenced at different altitudes [34,35]. Based on the overlap of differentially expressed genes among the oxygen-concentration comparison combinations, the yak group could be screened for 61 differentially expressed genes that were common or unique to yak. The overlap of differentially expressed genes among different combinations in the yellow cattle group resulted in the identification of 20 common or unique differentially expressed genes. This suggests that there are differences in the genes of species adapted to oxygen concentrations at different altitudes.

In GO analysis, the enriched genes in the biological process of PASMCs in the hypoxia 1 group vs. control were all shown to be upregulated, revealing that the most enriched entry was related to the DNA metabolism process, which was consistent with Bristow’s study [36], confirming that hypoxia may have the ability to repair DNA damage during DNA metabolism, allowing PASMCs to continue to proliferate. The results of KEGG analysis showed that DEGs common to both yaks and cattle were mainly enriched (*p* < 0.05) in the HIF-1 signaling pathway, carbon metabolism, glycolysis/gluconeogenesis, DNA replication, and other hypoxia-adaptive pathways. KEGG analysis showed that DEGs shared by yaks and cattle were mainly enriched (*p* < 0.05) in hypoxia-adaptive pathways such as the HIF-1 signaling pathway, carbon metabolism, glycolysis/gluconeogenesis, and DNA replication. The differential genes (HK2/PGK1/ENO1/ENO3/ALDOC/ALDOA/) were all upregulated in the hypoxia 1 vs. control group and were all downregulated in the hypoxia 10 vs. hypoxia 1 group, which was consistent with the results of RT-qPCR and RNA-Seq validation. The R^2^ of yellow cattle was measured as 0.9471, and the R^2^ of yaks was measured as 0.7635, which indicated that the correlation was better and the sequencing data were reliable. In the HIF-1-mediated adaptive response to promote cell survival under hypoxic conditions, there is an HIF-1-dependent reprogramming of glucose metabolism to reduce dependence on O_2_-dependent energy production. Under hypoxic conditions, increased glucose uptake and subsequent metabolism helps to maintain cellular ATP production while reducing oxidative metabolism [37]. HIF-1 binds to the hypoxia response element (HRE), which promotes an increase in its expression and mediates the upregulation of glucose transporter proteins and glycolytic enzymes [38]. HK2, one of the upregulated genes in this study, is the main rate−limiting enzyme in glycolysis, catalyzing the conversion of glucose to glucose-6-phosphate. It has been shown that HK2 not only activates autophagy under hypoxic stress conditions [39], but also that hypoxia upregulates HK2 in cells and induces its production of lactate, which increases the increased mRNA levels of glycolytic enzymes [40]. Therefore, it is hypothesized that hypoxic stress of yak and cattle PASMCs and the consequent upregulation of HK2 led to an increase in both autophagy and glycolytic processes in PASMCs, which, in turn, improved the quality of the PASMCs and thus adapted them to the hypoxic environment. PGK1 is the first key ATP-producing enzyme in the glycolytic pathway. PGK1 is not only a metabolic enzyme, but also a newly discovered protein kinase that coordinates glycolysis and mitochondrial metabolism [41]. PGK1 is an intracellular protein—an energy-producing glycolytic enzyme that catalyzes the reversible transfer of 1, 3-bisphosphoglycerate to ADP to produce 3-phosphoglycerate and ATP [41,42]. PGK1 not only plays an important role in the coordination of glycolysis and the tricarboxylic acid cycle, but also participates in the process of angiogenesis as a disulfide reductase [43]. Studies have shown that under hypoxic conditions, HIF-1α upregulates the expression of PGK1, which uses the glycolysis pathway to provide energy for the organism and participates in the angiogenesis process to meet the needs of the organism’s activities and then adapt to the external environment. This is similar to the results of the present study [44,45]. Therefore, it is concluded that PASMCs under hypoxic stress upregulate PGK1, which promotes glycolysis in PASMCs and is involved in angiogenesis. ALDOA aldolase A (ALDOA) and ALDOC aldolase C (ALDOC) are members of the aldolase family. The two isoforms are differentially expressed in tissues and are involved in glycolysis [46]. ALDOA has a role in glycolysis and plays an important role in maintaining glucose homeostasis [47]. Over the years, ALDOC has been found to be particularly abundant in brain tissue. It is present in normal brain tissue and is responsible for repairing damaged tissue [47,48,49]. In this study, both ALDOC and ALDOA were upregulated during hypoxia, which is consistent with [47]; it can be hypothesized that hypoxia induces the Warburg effect to acquire glycolysis and improve the survival of PASMCs by upregulating the expression of ALDOC and other related genes (*p* < 0.05). ALDOC was found to upregulate protein stability, disrupt ALDOC (which senses extracellular glucose deficiency), and promote sustained glucose uptake and glycolysis for cell growth [50]. KEGG analysis also showed that yak-unique DEGs were mainly enriched (*p* < 0.05) in the hypoxia-adaptive pathway with differential genes such as EHHADH, RGN, ALDH1A3, MCM7, POLD3, MCM6, PRIM1, RPA2, RFC4, and RFC3. These are linked to sugar metabolism pathways. The results of this study showed that the changes in differentially expressed genes in yak PASMCs under different oxygen concentrations were closely related to the molecular regulatory mechanism of hypoxia adaptation, which laid the foundation for further exploration of new genes related to hypoxia adaptation in yaks.

Yaks are endemic to the Tibetan Plateau and have a number of adaptations to the plateau region, such as strong pulmonary function, foraging ability, and metabolic capacity [1,2,3]. In this study, we found that the alveoli around the fine bronchioles at all levels of the adult yak lung were densely packed, whereas the alveoli around the fine bronchioles at all levels of the cattle lung were sparse and most of them were filled alveoli, whereas the alveoli of adult yaks showed the presence of both semi-filled and filled alveoli. Thus, we speculated that the unfilled alveoli were the semi-dormant and crumpled alveoli, the filled alveoli were restored only when the yaks were engaged in strenuous exercise, and this special structure greatly improved the respiratory−reserve capacity of yaks. This may be closely related to the adaptation of yaks to the hypoxic environment of the plateau and reduces water consumption during respiration. At the same time, studies have shown that abundant dilated capillaries and large numbers of erythrocytes in the alveolar septa are conducive to increased alveolar ventilation and oxygen transport [51,52,53,54,55,56]. We measured the percentage of alveolar septa at all levels of the peribronchiolar alveoli in yaks after H&E, Masson, and PAS staining, observing a highly significant difference compared to that of cattle, and we also observed that the alveolar septa in yaks between adjacent alveoli was rich in capillaries, and that the number of red blood cells was higher than that of cattle. The thickness of the alveolar septa was thicker than that of the cattle, and it was observed that the alveoli between adjacent alveoli were rich in capillaries and the number of red blood cells was higher than that of the cattle. This is one of the important mechanisms by which yaks adapt to the hypoxic environment of the plateau.

In addition, we combined the vascular criteria of animals written by Chen et al. [24] and found that the vascular type of fine bronchioles accompanying pulmonary arteries at all levels were pulmonary microarterioles in both cattle and yaks, and the mean thickness of the intima-media of fine bronchioles accompanying pulmonary arteries at all levels was significantly greater in yaks than in cattle, which was consistent with the results of Anand et al. [57] in a high-altitude study of yaks, yak-cattle crosses, and plains cattle. This is consistent with the results of Anand’s studies of high-altitude yaks, crosses between yaks and cattle, and domestic goats, suggesting that pulmonary microarterioles at all levels of fine bronchial arterioles are conducive to oxygen distribution, that the thicker intima-media of pulmonary arterioles of all levels of fine bronchial arterioles in the yak may be due to the low capacity of pulmonary vasoconstriction in hypoxic lungs after acclimatization, and that this trait is heritable.

Meanwhile, Masson’s staining showed that the fine bronchi and terminal bronchioles have the function of controlling the air flow into the alveoli, which is mainly achieved by diastole and contraction of the smooth muscle, and the respiratory fine bronchioles are closely connected with the alveoli and are an important pathway of oxygen exchange [58]. Therefore, the mean thickness of pulmonary smooth muscle at all levels of the fine bronchioles was observed and measured, and the mean thickness of smooth muscle at all levels of the fine bronchioles was found to be significantly greater in yaks than in cattle. The mean thickness of smooth muscle at all levels of pulmonary bronchioles in yaks was found to be significantly greater than that in cattle, and the proportion of collagen fibers at all levels of pulmonary bronchioles was found to be significantly higher in adult yaks than in cattle, suggesting that the high proportion of collagen fibers at all levels of pulmonary bronchioles in yaks may be related to better adaptation to the hypoxic environment. In addition, in combination with the results of Masson staining, it was observed that yaks have more elastic fibers, as well as dilated capillaries and a higher number of erythrocytes in the lumen of the tubes, all of which reflect one side of the argument that this structural change in yaks facilitates respiration and air–vascular and blood exchange and is beneficial for humidifying the air and removing secretions and other harmful substances [11,20,22]. This suggests that yaks, being native to the plateau, have undergone morphological changes adapted to the hypoxic environment of the plateau through a long period of natural selection.

The smooth muscle contractile capacity of the pulmonary artery vascular wall in yaks was stronger than that of the cattle, which was in favor of pulmonary artery blood flow, which is consistent with the structure of the study by Gao et al. [6]. This indicates that the pulmonary artery vascular wall in the lungs of yaks in the hypoxic environment of the plateau gradually adapts to its environment. The hypoxic environment may still cause thickening of the wall of the pulmonary arteries in yaks, leading to hyperplasia of the smooth muscle, but because it has already fully adapted to the hypoxia, yaks living in the hypoxic environment do not develop pathological changes such as pulmonary hypertension and right ventricular hypertrophy. In the intramural layer of the pulmonary arterial vascular system in the lungs of yaks, there was an increase in the amount of their smooth muscle without reduction and thinning, and this result may be the histological feature or the basis of the adaptation of the pulmonary arteries to hypoxia in yaks.

The results of PAS staining showed a positive PAS reaction of goblet cells in the fine bronchial epithelium of adult cattle and yaks, and it was also found that the percentage of goblet cells in the fine bronchioles of the lungs of adult yaks was high compared with that of cattle, suggesting that there is a large amount of glycogen in the goblet cells in the fine bronchial epithelium, which is conducive to mucous membrane protection [59]. In contrast, in the terminal fine bronchi and respiratory fine bronchi, the goblet cells disappeared, and the PAS reaction was negative. It was observed that the PAS reaction was positive in the peribronchial smooth muscle at all levels of the lung in both adult yaks and cattle, and the positive expression was stronger in the peribronchial smooth muscle at all levels of the lung in adult yaks than in cattle, indicating that the content of myoglycogen in the peribronchial smooth muscle at all levels of the lung was higher in yaks than in cattle, and the PAS reaction was positive in the peribronchial smooth muscle at all levels of the lung in adult yaks. The high distribution of glycogen in the peripheral smooth muscle of the fine bronchioles at all levels of the lung is conducive to providing energy for smooth muscle contraction and diastole during respiration in yaks.

The basement membrane (BM) is a specialized layer of extracellular matrix components that plays a central role in maintaining lung function [60]. Since the basement membrane plays a supporting role for tissue structures, we measured the thickness of the basement membrane of pulmonary fine bronchioles and terminal fine bronchioles and found that the thickness of the basement membrane in adult yaks was significantly greater than that in cattle, and it was hypothesized that the change in basement membrane thickness was associated with better adaptation to the hypoxic environment of the yak.

Immunohistochemical results showed that PDK1 was co-expressed with both HIF-1α and VEGF in the lungs of adult yaks and calves, suggesting that the regulation of PDK1 may be related to HIF-1α and VEGF. The expression of p-PDK1 was induced by HIF-1α to negatively regulate the entry of pyruvate into the TCA cycle [61] and promote PDK1 phosphorylation in mitochondria. p-PDK1 upregulates HIF-1α and promotes signal transduction, regulates the level of VEGF, and accelerates the process of vascular remodeling [62] to cope with plateau hypoxia or hypoxic environment, reflecting its extremely strong adaptive ability to plateau hypoxia.

The expression of HIF-1α and VEGF protein was stronger in yak lungs than in cattle lungs and was mainly expressed in terminal fine bronchial epithelial cells and alveolar septal cells in yak lungs and in terminal fine bronchial epithelial cells in cattle. Studies have shown that HIF-1α induces VEGF expression, and the protective effect of HIF-1α against hypoxia in Tibetan sheep lungs is achieved by inducing high VEGF expression, thereby ensuring vascular integrity [63]. Hypoxia induces VEGF expression in lung epithelial cells, and lung epithelial cells are involved in endothelial repair and angiogenesis after lung injury through VEGF synthesis [64]. As an important transcriptional regulator involved in the modulating angiogenesis, enhancement of vascular permeability, and vascular remodeling, the HIF-1α signaling pathway increases connective tissue expression in endothelial cells under hypoxic stimulation [65,66,67]. VEGF not only increases vascular permeability but also promotes the proliferation and differentiation of vascular smooth muscle cells, thereby promoting neovascularization [68,69,70]. Together with the experimental results, the structure of the lungs determines whether yaks can adapt to the hypoxic environment of the plateau [51,71,72]. The expression of HIF-1α and VEGF in the epithelial cells of terminal fine bronchioles of yaks is relatively high compared with that in cattle, which is conducive to the protection of bronchioles and the enrichment of the soft tube mechanism in yaks under the hypoxic environment of the plateau and facilitates the contraction of bronchioles. In addition, the pulmonary vasculature of yaks has a thinner wall and a relatively low content of smooth muscle, which is a good indication that the pulmonary vasculature is suitable for vascular contraction and adequate pressure to transport blood and maintain its ability to respond to the hypoxic environment. HIF-1α and VEGF expressions were higher in yaks than in cattle, indicating that the structure of yak pulmonary vasculature is favorable for vasoconstriction under the hypoxic environment of the plateau. As a result, the vasculature can produce enough pressure to transport blood and maintain its adaptation to the hypoxic environment, whereas cattle live in the oxygen-rich areas of the plains, so that the expressions of HIF-1α and VEGF were lower. It is suggested that these three factors regulate metabolic pathways, vascular permeability, and energy circulation in yaks during hypoxic adaptation but are not involved in the regulation of vascular proliferation, vascular remodeling, etc. Further studies are needed to investigate the intracellular regulation and hypoxia adaptation of PDK1, HIF-1α, and VEGF in yaks.

## 5. Conclusions

In summary, in the present study, we used RNA-Seq to explore the differences in hypoxic adaptation between yaks and cattle PASMCs under different oxygen concentrations and the differences in lung tissue structure and expression of related factors between yaks and cattle, and found that the adaptation of yak PASMCs to the high-altitude environment may be mediated through the regulation of the molecular mechanisms of hypoxic acclimatization in these pathways, such as the HIF-1 signaling pathway and glucose metabolism pathway. It was found that the process of adaptation to high-altitude environments in yak PASMCs may be regulated by the molecular mechanism of hypoxia adaptation through the HIF-1 signaling pathway and glucose metabolism pathway, and that the differentially expressed genes, such as HK2/PGK1/ENO1/ENO3/ALDOC/ALDOA, are closely related to the mechanism of hypoxia molecular regulation in these pathways. Compared with cattle lung tissues, yak lungs had semi-filled alveoli around fine bronchioles at all levels, thicker alveolar septa, a higher percentage of collagen fibers, a greater number of erythrocytes, greater distribution of capillaries, etc. There was a higher percentage of goblet cells in the fine bronchioles of the yak, and the relative expression of HIF-1α and VEGF mRNA was significantly higher in the lungs of adult yaks compared with that of cattle (*p* < 0.05). The relative expression of PDK1 mRNA was not significant, the relative expression of PDK1 and HIF-1α protein in the lungs of adult yaks was significant (*p* < 0.01), and the relative expression of PDK1 protein was not significant. These results constitute further data regarding the regulatory relationship between PDK1, HIF-1α, and VEGF with respect to the regulation of hypoxia acclimatization as well as the mechanism underlying yak plateau acclimatization, providing basic information and target for the prevention and treatment of plateau diseases in humans and plateau animals.

## Figures and Tables

**Figure 1 animals-13-03450-f001:**
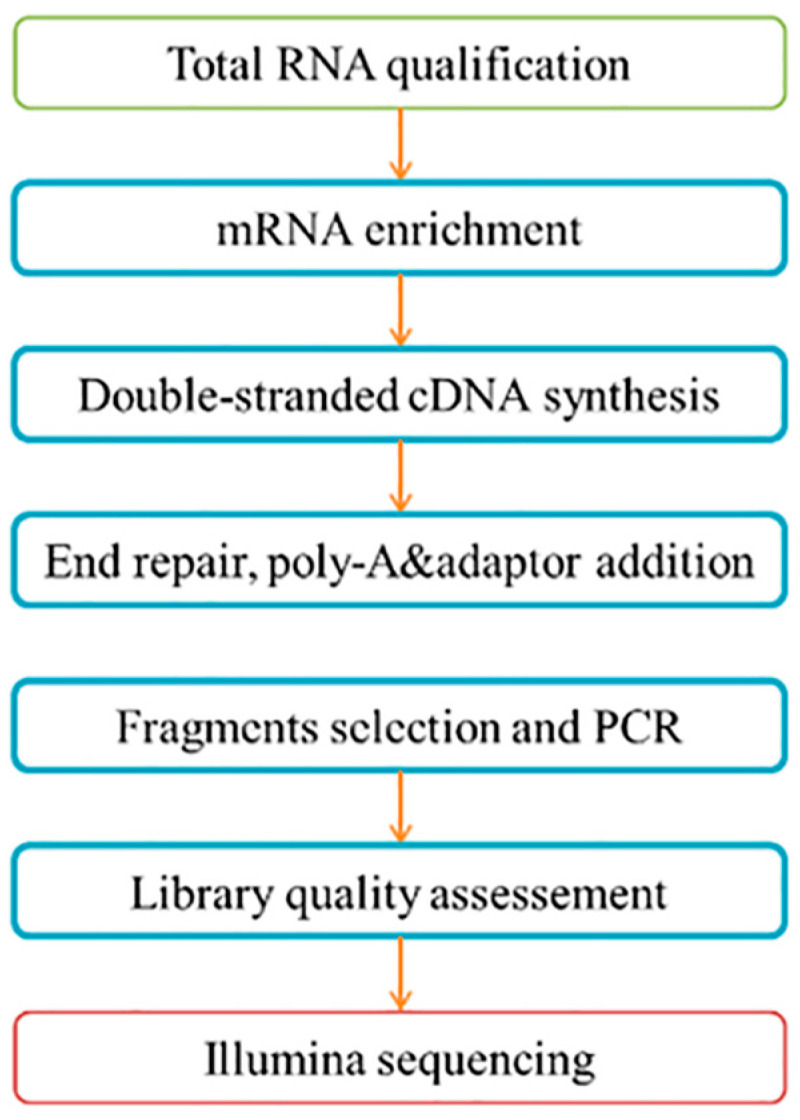
The flow of sequencing and database construction.

**Figure 2 animals-13-03450-f002:**
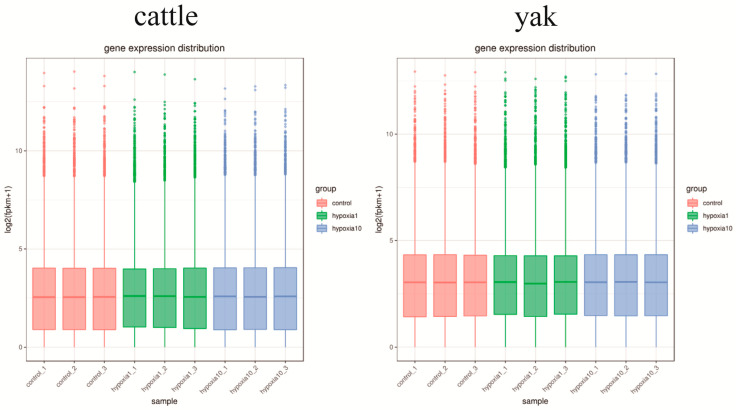
Box diagram of gene expression level distribution in samples. In the figure, the abscissa is the sample name, and the ordinate is log2 (FPKM + 1). Groupings in the figure are as follows: control refers to 20% O_2_ concentration; hypoxia 1 refers to 1% O_2_ concentration; hypoxia 10 refers to 10% O_2_ concentration; FPKM indicates fragments per thousand transcribed bases per million mapped reads.

**Figure 3 animals-13-03450-f003:**
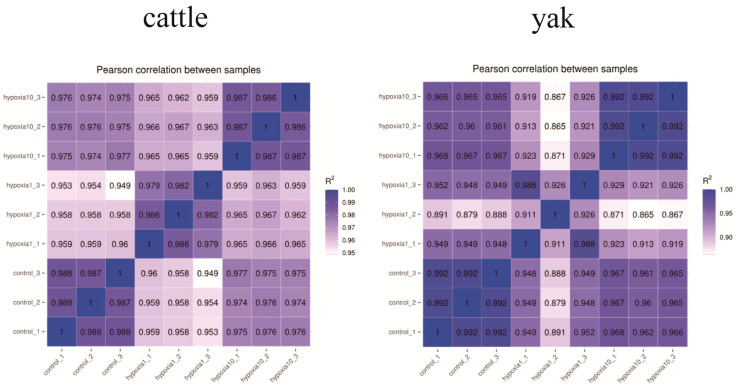
Heat map of correlation between samples. Groupings in the figure are as follows: control refers to 20% O_2_ concentration; hypoxia 1 refers to 1% O_2_ concentration; hypoxia 10 refers to 10% O_2_ concentration. The _1, _2, and _3 tails were named for each of the three samples in each group.

**Figure 4 animals-13-03450-f004:**
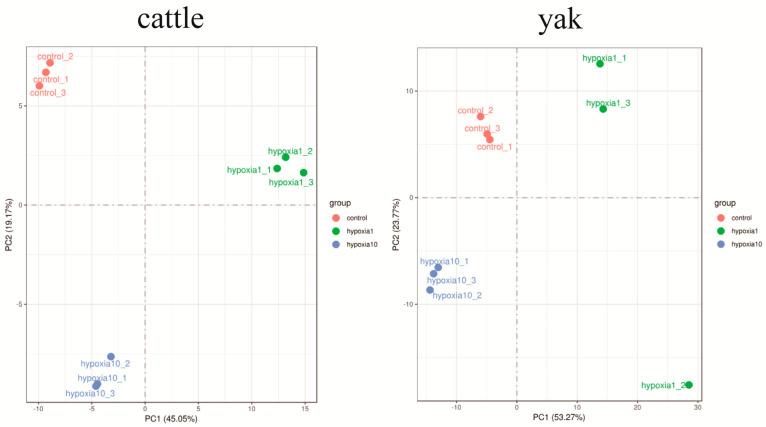
Principal component analysis (PCA). Groupings in the figure are as follows: control refers to 20% O_2_ concentration; hypoxia 1 refers to 1% O_2_ concentration; hypoxia 10 refers to 10% O_2_ concentration; the _1, _2, and _3 tails were named for each of the three samples in each group.

**Figure 5 animals-13-03450-f005:**
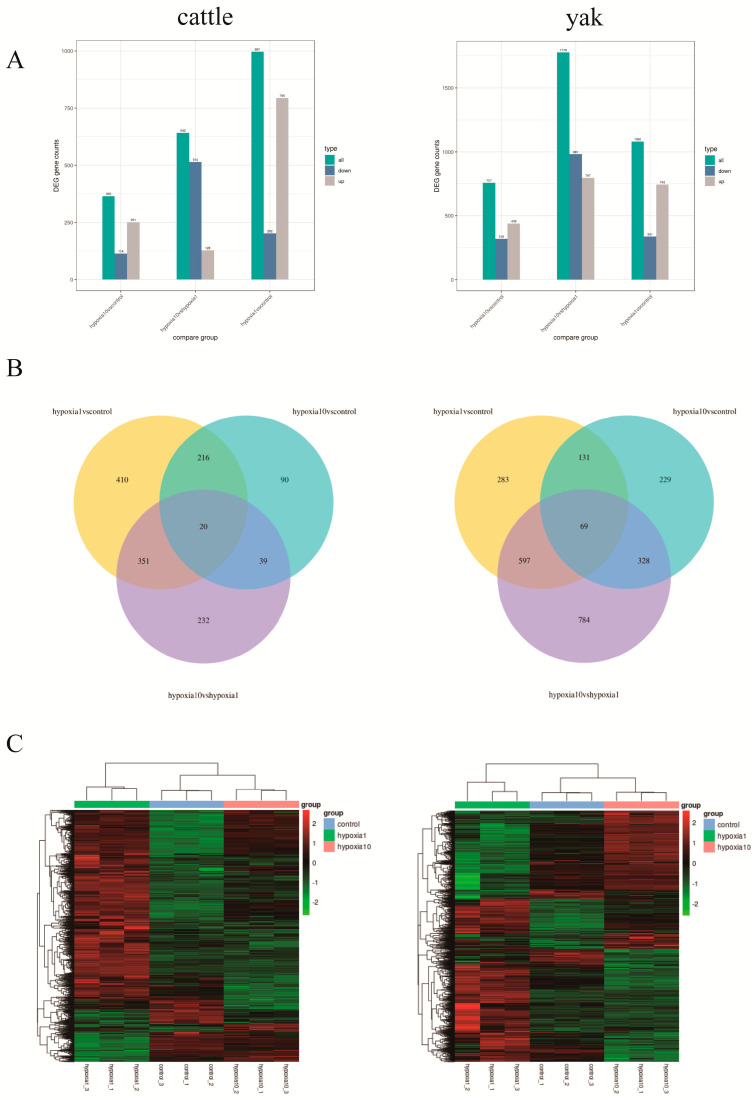
Transcriptome analysis of yak and cattle PASMCs under different oxygen concentration conditions. (**A**) Statistical histogram of the number of differential genes in the differential comparison combinations; blue and gray indicate the upregulated and downregulated differential genes, respectively, and the numbers on the bars indicate the number of differential genes; (**B**) Wayne diagram of the differential genes; (**C**) heat map of the clustering of the differentially expressed genes; the horizontal coordinate is the name of the samples, and the vertical coordinate is the normalized FPKM value of the differential genes.

**Figure 6 animals-13-03450-f006:**
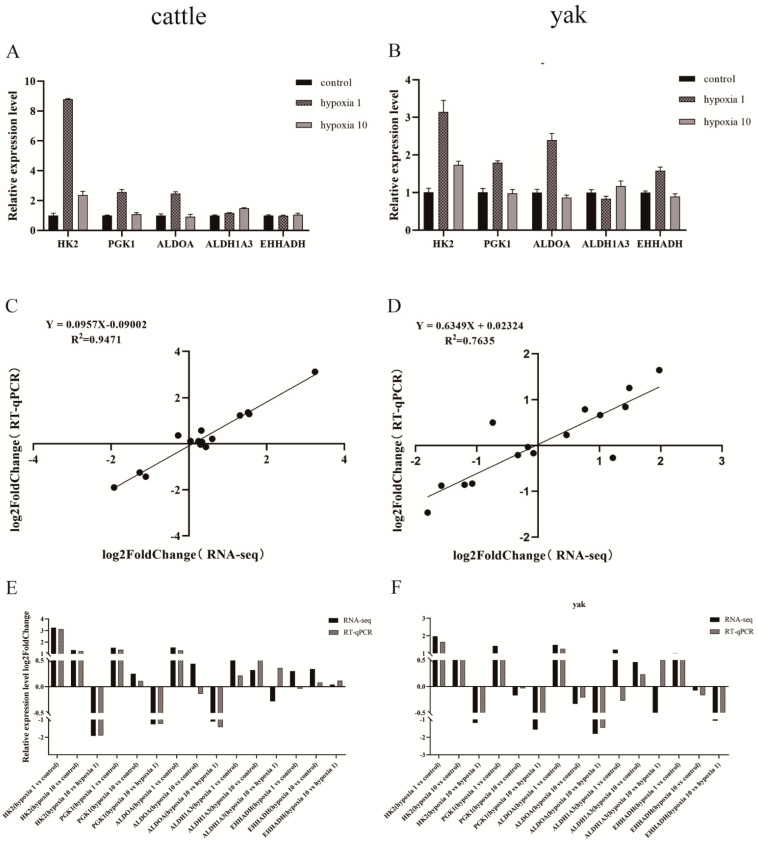
Expression patterns of differentially expressed genes. (**A**) Transcriptome sequencing was verified by RT-qPCR in cattle PASMCs; (**B**) transcriptome sequencing was verified by RT-qPCR in yak PASMCs; (**C**) correlation between RNA-Seq and RT-qPCR in cattle PASMCs; (**D**) correlation between RNA-Seq and RT-qPCR in yak PASMCs; (**E**) expression patterns of differentially expressed genes by RT-qPCR and RNA-Seq in cattle PASMCs; (**F**) expression patterns of differentially expressed genes by RT-qPCR and RNA-Seq in yak PASMCs. Groupings in the figure: control refers to 20% O_2_ concentration; hypoxia 1 refers to 1% O_2_ concentration; hypoxia 10 refers to 10% O_2_ concentration.

**Figure 7 animals-13-03450-f007:**
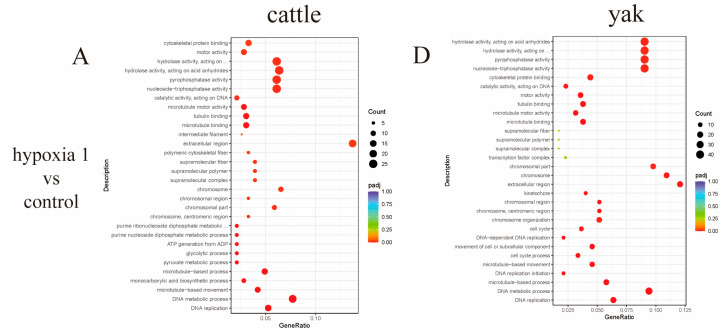
Scatterplot of GO enrichment analysis under gradient oxygen concentration. The horizontal coordinate of the graph is the ratio of the number of differential genes annotated with the GO term to the total number of differential genes, and the vertical coordinate is the GO term. (**A**–**C**) Scatterplot of GO enrichment analysis of cattle PASMCs under hypoxia 1 vs. control, hypoxia 10 vs. control, and hypoxia 10 vs. hypoxia 1; (**D**–**F**) scatterplot of GO enrichment analysis of yak PASMCs under hypoxia 1 vs. control, hypoxia 10 vs. control, and hypoxia 10 vs. hypoxia 1. The omitted part of the figure is: hydrolase activity, acting on acid anhydrides, in phosphorus-containing anhydrides; purine ribonucleoside diphosphate metabolic process; purine ribonucleoside diphosphate metabolic process; oxidoreductase activity, acting on single donors with incorporation of molecular oxygen; hydrolase activity, acting on acid anhydrides.

**Figure 8 animals-13-03450-f008:**
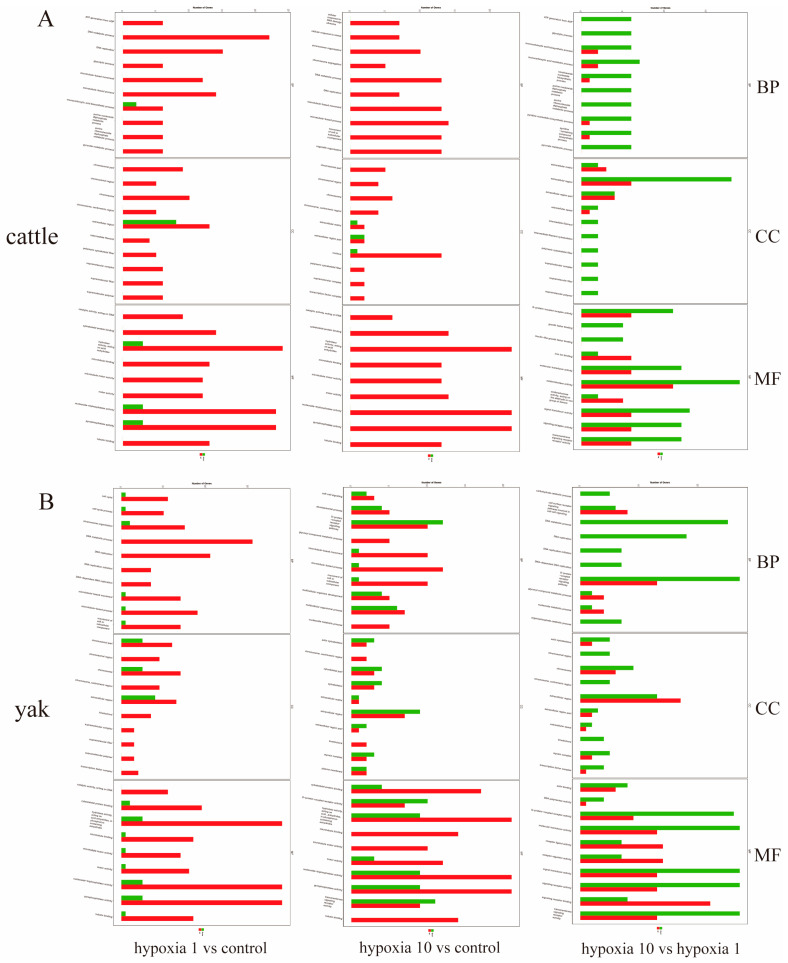
Histogram of GO enrichment analysis under gradient oxygen concentration. The horizontal coordinate of the graph is the number of genes enriched by GO term, and the vertical coordinate is the number of GO terms (BP: biological process, MF: molecular function, CC: cellular component). Groupings in the figure are as follows: control refers to 20% O_2_ concentration; hypoxia 1 refers to 1% O_2_ concentration; hypoxia 10 refers to 10% O_2_ concentration. (**A**) Histogram of GO enrichment analysis of cattle PASMCs under hypoxia 1 vs. control, hypoxia 10 vs. control, and hypoxia 10 vs. hypoxia 1; (**B**) histogram of GO enrichment analysis of yak PASMCs under hypoxia 1 vs. control, hypoxia 10 vs. control, and hypoxia 10 vs. hypoxia 1.

**Figure 9 animals-13-03450-f009:**
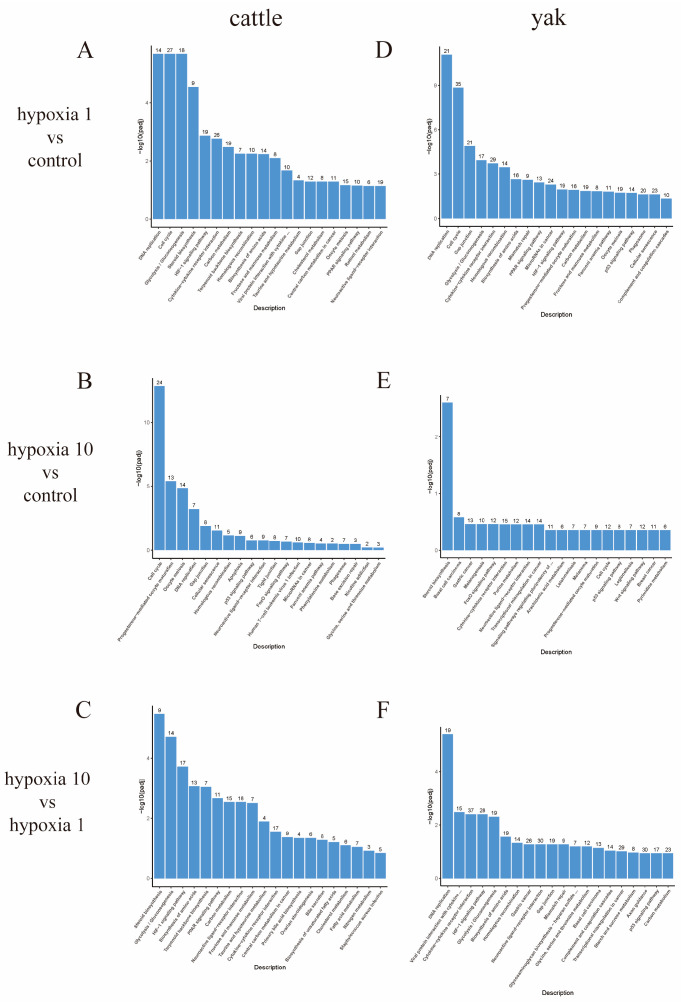
Bar graph of KEGG enrichment analysis. The horizontal coordinate of the graph is the KEGG pathway and the vertical coordinate is the significance level of the pathway enrichment. Groupings in the figure are as follows: control refers to 20% O_2_ concentration; hypoxia 1 refers to 1% O_2_ concentration; hypoxia 10 refers to 10% O_2_ concentration. (**A**–**C**) Bar graph of KEGG enrichment analysis of cattle PASMCs under hypoxia 1 vs. control, hypoxia 10 vs. control, and hypoxia 10 vs. hypoxia 1; (**D**–**F**) bar graph of KEGG enrichment analysis of yak PASMCs under hypoxia 1 vs. control, hypoxia 10 vs. control, and hypoxia 10 vs. hypoxia 1. The omitted part of the figure is: Viral protein interaction with cytokine and cytokine receptor; Signaling pathways regulating pluripotency of stem cells.

**Figure 10 animals-13-03450-f010:**
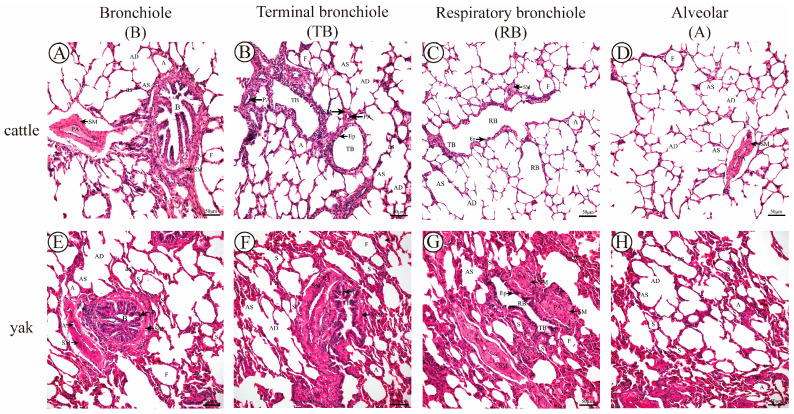
H&E staining of adult cattle and yak lung tissue. (**A**–**D**), respectively, correspond to PAS staining of lung bronchioles (B), terminal bronchioles (TB), respiratory bronchioles (RB), and alveoli (A) in adult cattle, 200×; (**E**–**H**), respectively, correspond to H&E staining of lung bronchioles (B), terminal bronchioles (TB), respiratory bronchioles (RB), and alveoli (A) in adult yaks, 200×. B: bronchiole; TB: terminal bronchiole; RB: respiratory bronchiole; PA: pulmonary artery; A: alveoli; F: filled alveoli; S: semi−filled alveoli; AD: alveolar duct; AS: alveolar sac; as: alveolar septa; Ep: epithelium cell; and SM: smooth muscle.

**Figure 11 animals-13-03450-f011:**
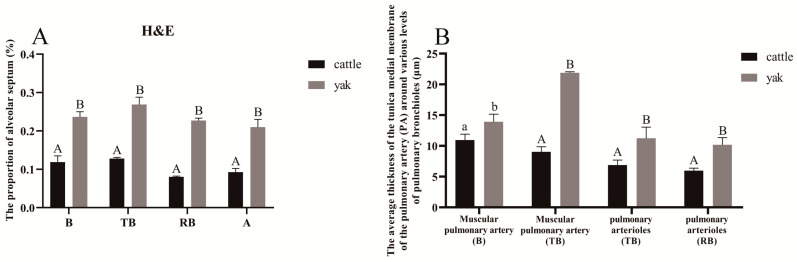
Analysis of data from H&E measurements of fine bronchioles at all levels of the lungs in adult cattle and yaks. (**A**) The proportion of alveolar septa in H&E staining of various levels of pulmonary bronchioles in adult cattle and yaks; (**B**) the average thickness of the accompanying pulmonary artery tunica medial at various levels of the pulmonary bronchioles in adult cattle and yaks. Different letters indicate significant differences, capital letters indicate extremely significant differences (*p* < 0.01), lowercase letters indicate significant differences (*p* < 0.05), and the same letters indicate no significant difference (*p* > 0.05); *n* = 3.

**Figure 12 animals-13-03450-f012:**
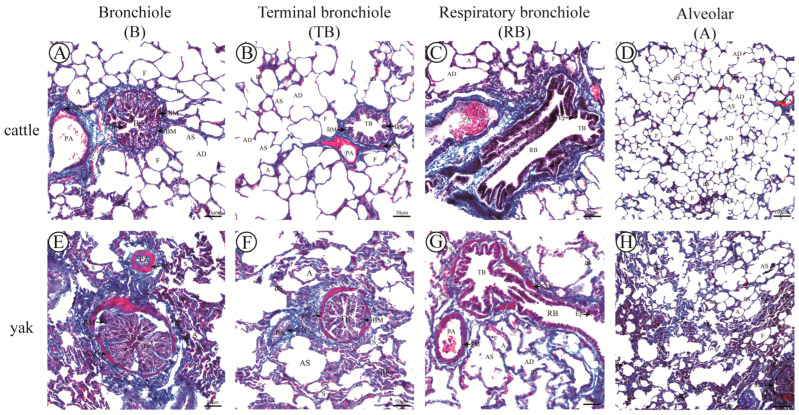
Masson staining of adult cattle and yak lung tissue. (**A**–**C**), respectively, correspond to Masson staining of lung bronchioles (B), terminal bronchioles (TB), and respiratory bronchioles (RB) in adult cattle, 200×; (**E**–**G**), respectively, corresponding to Masson staining of lung bronchioles (B), terminal bronchioles (TB), and respiratory bronchioles (RB) in adult yaks, 200×; (**D**,**H**), respectively, correspond to Masson staining of alveoli (A) in adult cattle and yak, 100×. B: bronchiole; TB: terminal bronchiole; RB: respiratory bronchiole; PA: pulmonary artery; A: alveoli; F: filled alveoli; S: semi−filled alveoli; AD: alveolar duct; AS: alveolar sac; as: alveolar septa; Ep: epithelium cell; GC: goblet cell; SM: smooth muscle; and BM: basement membrane.

**Figure 13 animals-13-03450-f013:**
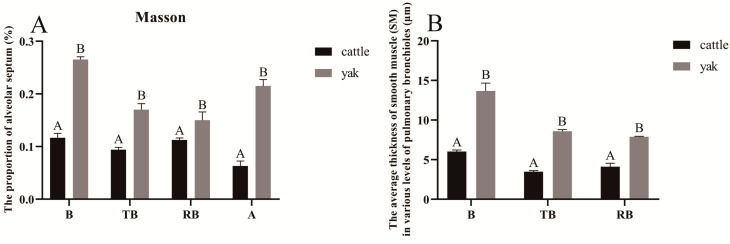
Analysis of data from Masson measurements of fine bronchioles at all levels of the lungs in adult cattle and yaks. (**A**) The proportion of alveolar septa in Masson staining of various levels of pulmonary bronchioles in adult cattle and yaks; (**B**) the average thickness of smooth muscle in various levels of pulmonary bronchioles in adult cattle and yaks; (**C**) the proportion of collagen fibers in various levels of pulmonary bronchioles in adult cattle and yaks. Different letters indicate significant differences, capital letters indicate extremely significant differences (*p* < 0.01), lowercase letters indicate significant differences (*p* < 0.05), and the same letters indicate no significant difference (*p* > 0.05); *n* = 3.

**Figure 14 animals-13-03450-f014:**
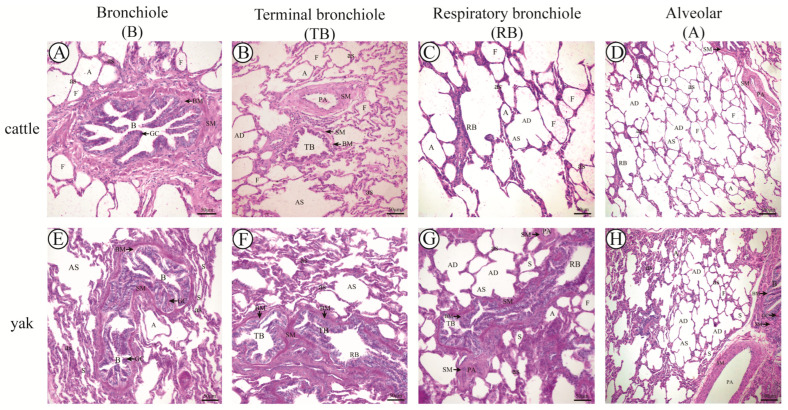
PAS staining of adult cattle and yak lung tissue. (**A**–**C**), respectively, correspond to PAS staining of lung bronchioles (B), terminal bronchioles (TB), and respiratory bronchioles (RB) in adult cattle, 200×; (**E**–**G**), respectively, corresponding to PAS staining of lung bronchioles (B), terminal bronchioles (TB), and respiratory bronchioles (RB) in adult yaks, 200×; (**D**,**H**), respectively, correspond to PAS staining of alveoli (A) in adult cattle and yaks, 100×. B: bronchiole; TB: terminal bronchiole; RB: respiratory bronchiole; PA: pulmonary artery; A: alveoli; F: filled alveoli; S: semi−filled alveoli; AD: alveolar duct; AS: alveolar sac; as: alveolar septa; Ep: epithelium cell; GC: goblet cell; SM: smooth muscle; and BM: basement membrane.

**Figure 15 animals-13-03450-f015:**
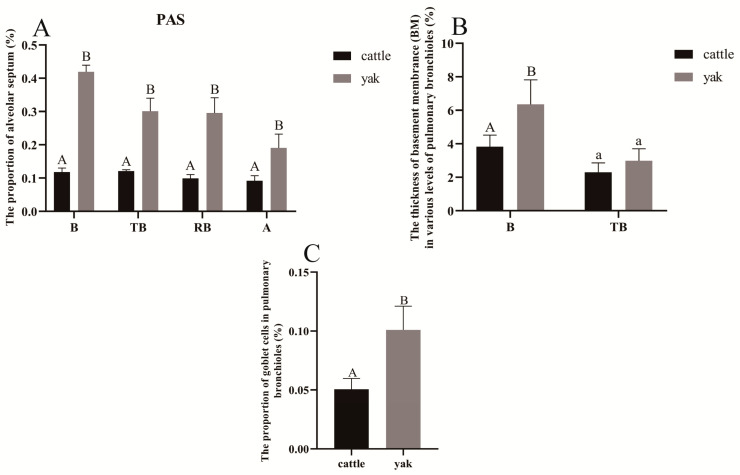
Analysis of data from PAS measurements of fine bronchioles at all levels of the lungs in adult cattle and yaks. (**A**) The proportion of alveolar septa in PAS staining of various levels of pulmonary bronchioles in adult cattle and yaks; (**B**) the thickness of basement membrane in various levels of pulmonary bronchioles in adult cattle and yaks; (**C**) the proportion of goblet cells in pulmonary bronchioles of adult cattle and yaks. Different letters indicate significant differences, capital letters indicate extremely significant differences (*p* < 0.01), lowercase letters indicate significant differences (*p* < 0.05), and the same letters indicate no significant difference (*p* > 0.05); *n* = 3.

**Figure 16 animals-13-03450-f016:**
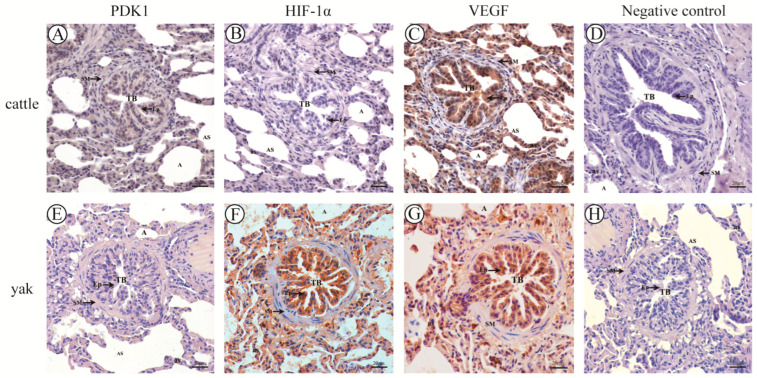
Distribution of PDK1, HIF-1α, and VEGF in adult cattle and yak lung tissue. (**A**–**C**), respectively, correspond to the immunohistochemistry of adult cattle lung tissue PDK1, HIF-1α, and VEGF; (**E**–**G**) correspond to the immunohistochemistry of adult yak lung tissue PDK1, HIF-1α, and VEGF, respectively; (**D**–**H**), respectively, correspond to the negative control of adult cattle and yak lung tissue; 400×. TB: terminal bronchiole; PA: pulmonary artery; AS: alveolar sac; as: alveolar septa; Ep: simple columnar epithelium; and SM: smooth muscle.

**Figure 17 animals-13-03450-f017:**
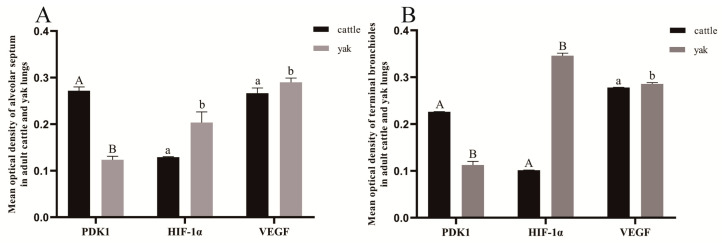
Mean optical density values of PDK1, HIF-1α, and VEGF in adult cattle and yak lung tissue. (**A**) The mean optical density value of PDK1 and its related factors in the terminal bronchioles of adult cattle and yaks; (**B**) the mean optical density of PDK1 and its related factors in the alveolar septa of adult cattle and yaks. Different letters indicate significant differences, capital letters indicate extremely significant differences (*p* < 0.01), lowercase letters indicate significant differences (*p* < 0.05), and the same letters indicate no significant difference (*p* > 0.05); *n* = 3.

**Figure 18 animals-13-03450-f018:**
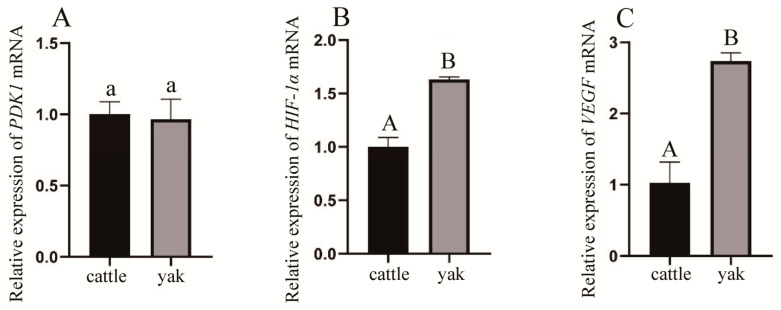
mRNA expression of PDK1, HIF-1α, and VEGF in adult cattle and yak lung tissue. (**A**–**C**) The relative expression levels of PDK1, HIF-1α, and VEGF mRNA in adult cattle and yak lungs, respectively. Different letters indicated significant difference (*p* < 0.05), same letters indicate no significant difference (*p* > 0.05), capital letters indicate extremely significant differences (*p* < 0.01), lowercase letters indicate significant differences (*p* < 0.05), and the same letters indicate no significant difference (*p* > 0.05); *n* = 3.

**Figure 19 animals-13-03450-f019:**
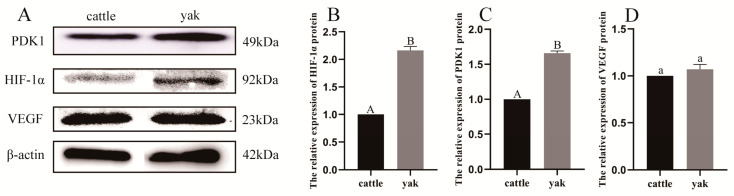
Protein expression of PDK1, HIF-1α, and VEGF in adult yak and cattle lungs. (**A**) The expressions of PDK1, HIF-1α, and VEGF proteins in the lungs of adult cattle and yaks. (**B**–**D**) Relative expression of PDK1, HIF-1α, and VEGF protein. Different letters indicate significant differences, capital letters indicate extremely significant differences (*p* < 0.01), lowercase letters indicate significant differences (*p* < 0.05), and the same letters indicate no significant difference (*p* > 0.05).

**Table 1 animals-13-03450-t001:** Primer sequence information.

Primer	Primer Sequence(5′~3′)	Fragment Size/bp
HK2	F: GAGTTGGCAGGATGATTG	190
	R: AGAAAGACGCATGTGGTA	
PGK1	F: GCTGACAAGAATGGCGTGAA	182
	R: TGCTTAGCCCGAGCAACA	
ALDOA	F: CACCAATACCCAGCACTCAC	181
	R: GCAGTTGGCGGTAGAAGC	
ALDH1A3	F: TGGAGTATGCCAAGAAGCG	251
	R: TGGTTGCACTGGTCCAAAA	
EHHADH	F: TCCTCTGTTGGCGTTCTC	137
	R: ATGGAGGTTATTATCTTGCTTG	
PDK1	F: AGTATGGCTCAAAGCTGCCC	106
	R: ACAACTTAAGTCTCGCGGCA	
HIF-1α	F: GGCGCGAACGACAAGAAAAA	121
	R: GTGGCAACTGATGAGCAAGC	
VEGF	F: CTGCTGTGGACTTGAGTTGGG	107
	R: GCTGCCGTAAGAGGGATAAAA	
β-actin (ACTB)	F: TCATCACCATCGGCAATGAG	157
	R: AGCACCGTGTTGGCGTAGAG	

Note: HK2, hexokinase 2; PGK1, phosphoglycerate kinase 1; ALDOA, aldolase A, fructose bisphosphate; ALDH1A3, aldehyde dehydrogenase 1 family, member A3; EHHADH, enoyl-Coenzyme A, hydratase/3-hydroxyacyl Coenzyme A dehydrogenase; PDK1, pyruvate dehydrogenase kinase 1; HIF-1α, hypoxia inducible factor 1α; VEGF, vascular endothelial growth factor; β-actin (ACTB), actin beta.

**Table 2 animals-13-03450-t002:** Standard for classification of arterial vascular types [24].

Artery Type	Vascular Diameter Size (mm)
Elastic artery	about 150
Muscular artery	1–20
Small artery	0.3–1
Pulmonary arterioles	0.03–0.3

**Table 3 animals-13-03450-t003:** Pulmonary artery diameter of various levels of bronchioles in cattle and yaks.

Bronchioles of All Levels	Accompanying Pulmonary Artery Diameter in Cattle (mm)	Accompanying Pulmonary Artery Diameter in Yaks (mm)
Bronchiole	0.1691	0.1905
Terminal bronchiole	0.0338	0.0432
Terminal bronchiole	0.1291	0.1939
Respiratory bronchioles	0.0321	0.0728

## Data Availability

All data presented in this study are available on request from the corresponding authors.

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
