# Peer review of "Study of Transcriptomic Analysis of Yak (Bos grunniens) and Cattle (Bos taurus) Pulmonary Artery Smooth Muscle Cells under Oxygen Concentration Gradients and Differences in Their Lung Histology and Expression of Pyruvate Dehydrogenase Kinase 1-Related Factors"

_animals, 2023, doi:10.3390/ani13223450_

Round 1

Reviewer 1 Report

Comments and Suggestions for Authors

Dear Authors,

Here are my interpretations and suggestions for this article:

This article analyses the hypoxic adaptation of the yak from both in vivo and in vitro perspectives, combining transcriptomic analysis, molecular experiments, cellular and tissue levels, and thoroughly analyses the hypoxic adaptation of the yak from a comparative perspective of yak and yellow bovine, and its findings of the HIF-1 signaling pathway as well as glucose metabolism and other related pathways and factors, PDK1, HIF-1α and VEGF as entry points. The in vivo study was conducted and found that there are differences and the regulatory relationship between the three may be related to the hypoxic adaptation of yaks, which is expected to provide target genes for the treatment of pulmonary hypertension and other plateau diseases. I believe that the study is innovative and novel and that the article deserves to be published with minor revisions.

1. About the manuscript of the manuscript: In this manuscript, only yak and cattle pulmonary artery smooth muscle cells were exposed to different oxygen concentration gradients, while yak and cattle lung tissues were not, so the order of the statements in the title of the manuscript needs to be adjusted. It is suggested that the phrase "Under the Oxygen Concentration Gradient" be changed to "Yak (Bos grunniens) and Cattle (Bos taurus) PASMCs". Specific changes have been made according to the content of the manuscript.

2. Regarding the Simple Summary and Abstract sections: the tense of "prolonged" in line 23 of the manuscript is inconsistent with the tense of this sentence, so please use "prolong"; change "paper" to "manuscript" in lines 26 and 38. Please change "paper" to "manuscript" in lines 26 and 38.

3. Regarding the Materials and Methods section, please revise the unit "cm3" to "cm3" in lines 115, 116, and 170 of the manuscript.

4. Regarding the Results section: please revise "O2" to "O2" in lines 293-295 and 300-303 in section 3.4 of the manuscript; and in the Immunohistochemistry section, Figure 16, Figure H (Yak negative control). The quality of the image is poor and needs to be revised.

5. Regarding the Discussion section: Please summarize and discuss the paragraph on line 660 for the Transcriptome Analysis paragraph.

Good luck,

September 8th, 2023

Comments on the Quality of English Language

Dear Authors,

(1) After reading the entire article, I think the author's overall focus is on "cattle". In lines 86, 89, 110, 118, 238, 267, 409, 552 and 791 of the article, there are other terms such as "yellow cattle" and "buffalo", so please standardize the whole article to "cattle" to make it easier to understand;

(2) There are instances in the manuscript where both British and American style English are present, please harmonize.

Good luck,

September 8th, 2023

Author Response

Dear Reviewer:

Greetings! Thank you for taking the time to review the manuscript (animals-2608904). Based on your valuable comments, I have revised the manuscript according to your comments, and the specific revisions are as follows:

  1. About the manuscript of the manuscript: In this manuscript, only yak and cattle pulmonary artery smooth muscle cells were exposed to different oxygen concentration gradients, while yak and cattle lung tissues were not, so the order of the statements in the title of the manuscript needs to be adjusted. It is suggested that the phrase "Under the Oxygen Concentration Gradient" be changed to "Yak (Bos grunniens) and Cattle (Bos taurus) PASMCs". Specific changes have been made according to the content of the manuscript.

Answer: Thank you for the valuable suggestions from the experts, as per your suggestion, The title has been revised and changed in the revised manuscript to " Study of Transcriptomic Analysis of Yak (Bos grunniens) and Cattle (Bos taurus) PASMCs Under Oxygen Concentration Gradients and Differences in Their Lung Histology and Expression of PDK1- Related Factors".

  1. Regarding the Simple Summary and Abstract sections: the tense of "prolonged" in line 23 of the manuscript is inconsistent with the tense of this sentence, so please use "prolong"; change "paper" to "manuscript" in lines 26 and 38. Please change "paper" to "manuscript" in lines 26 and 38.

Answer: Thank you for the valuable expert advice, which has been incorporated into the revised manuscript in accordance with your suggestions.

  1. Regarding the Materials and Methods section, please revise the unit "cm3" to "cm3" in lines 115, 116, and 170 of the manuscript.

Answer: Thank you for the valuable expert advice, which has been incorporated into the revised manuscript in accordance with your suggestions.

  1. Regarding the Results section: please revise "O2" to "O2" in lines 293-295 and 300-303 in section 3.4 of the manuscript; and in the Immunohistochemistry section, Figure 16, Figure H (Yak negative control). The quality of the image is poor and needs to be revised.

Answer: Thank you for the valuable advice from the reviewer, which has been changed in the revised manuscript according to your suggestions. See Figure 16, line 590 of the revised manuscript for details.

  1. Regarding the Discussion section: Please summarize and discuss the paragraph on line 660 for the Transcriptome Analysis paragraph.

Answer: Thank you for the valuable advice from the experts, in accordance with your suggestions, the revision of the manuscript has been modified to add the following content “The results of this study showed that the changes of differentially expressed genes in yak PASMCs under different oxygen concentrations were closely related to the mo-lecular regulatory mechanism of hypoxia adaptation, which laid the foundation for further exploration of new genes related to hypoxia adaptation in yaks“. See lines 719-723 of the revised manuscript for details.

  1. After reading the entire article, I think the author's overall focus is on "cattle". In lines 86, 89, 110, 118, 238, 267, 409, 552 and 791 of the article, there are other terms such as "yellow cattle" and "buffalo", so please standardize the whole article to "cattle" to make it easier to understand;

Answer: Thank you for the valuable advice of experts, in accordance with your suggestions, in the revision of the manuscript has been revised.

  1. There are instances in the manuscript where both British and American style English are present, please harmonize.

Answer: Thank you for your valuable suggestions, as per your suggestions, this manuscript has now been edited and proofread in English, please see the revised manuscript for more details.

Finally, I would like to thank the experts for their comments and suggestions that improved this manuscript and for your recognition of this article.

Looking forward to your reply. 

Kind regards,

Mr. Zhang,

E-Mail: [email protected]

Reviewer 2 Report

Comments and Suggestions for Authors

Summary:

The manuscript examines lungs of cattle from 1000 m above sea level and yak from 3500 m above sea level histologically and by RNA sequencing. Authors have also examined pulmonary artery smooth muscle cells cultured invitro under different oxygen concentrations mimicking hypoxia and have examined gene expression differences using RNA sequencing technology and have analyzed expression patterns of selected pathways known to play a role in hypoxic environments. They identified several genes differentially regulated in yak and cattle and under hypoxic conditions.

Review:

Although this manuscript covers an interesting and potentially useful topic, it is incomplete with data and figures missing and written extremely poorly. Some of the experiments described are redundant in nature and data and methodology for seminal experiments described in the manuscript is missing. Terminology used in the text and in figures are different making it difficult to interpret results and discussion. It seemed as if no one has proofread the manuscript prior to submission.

The central experiment described in the manuscript is in-vitro cutlure of cattle/yak lung tissue under different oxygen concentrations mimicking hypoxia. The method that experiment was conducted is missing in the methods section making it hard to evaluate the results. Further, all figures use the terminology Hypoxia 1, Hypoxia 10 etc. The word is not found anywhere in the text of the manuscript.

Multiple staining methods take up a lot of the manuscript, however, the reason for using different staining methods to prove the same effect is redundant and not justified.

Figure 2 is missing. It is one of the most important figures setting up later experiments and it looks like a whole page (or more) of information is missing from the document provided for review. (line numbers, however, correlate, proving that pages were missing before line numbers were inserted).

Figure 16 is missing although the beginning of the figure legend is present as well as several references to the figure in the text. (It could be that the figure intended to be 16 is indeed figure 17. However, reviewers should not be expected to make that determination)

Most of the discussion is not supported by the data presented.

This manuscript needs extensive language editing and careful proof-reading before it is allowed to be reviewed again.

Specific comments:

Line 59-60 – The sentence needs to be edited for clarity.

Line 68-69 – The sentence needs to be edited for clarity/grammar

Line 76 – What is PAH2? It is a known gene that has no relevance to hypoxia.

Line 104-108 – Needed to be edited

Line 129 – “junctions” did you mean adaptors?

Line 138 – the second DESq2 part of the sentence is a duplicate.

Line 143 – Need a reference for the sentence ending with “rates”

Line 263 – Define FPKM

Lin 269 – Where is the figure?

Line 270 – data/text missing from here

Figure 3 – What is Hypoxia 1 and 2

Line 296 (and elsewhere) O2 should be O2

Figure 6 – Figure and the text uses different nomenclature.

Table 2 – are these numbers corrected for the size of the animal?

Line 544 – Is this figure meant to be Figure 16

Line 590-592 – Sentence needs to be edited.

Line 613-617 – your data does not support this statement.

Line 804-821 – Placeholder text provided by the journal is still present in this section.

Comments on the Quality of English Language

The manuscript needs extensive English editing as well as proofreading. There are figure missing, spelling errors and even some placeholder text provided by the journal is present in the manuscript. 

Author Response

Dear Reviewer:

Greetings! Thank you for taking the time to review the manuscript (animals-2608904). Based on your valuable comments, I have revised the manuscript according to your comments. Please see the attachment.

Kind regards,

Mr. Zhang,

October 27th, 2023

Reviewer 3 Report

Comments and Suggestions for Authors

In this manuscript, The molecular mechanisms by which hypoxia affects the biological behavior of yak PASMCs and the changes in the histological structure of yak and cattle lungs were investigated. Minor revision should be performed.

Specific comments:

(1) Please add more absolute numbers like the parameter combination values etc. to the abstract and conclusions sections to improve its information content rather than state vague trends.

(2) The introduction section could be extended to clarify the novelty of this manuscript further. More literatures are needed.

Author Response

Dear Reviewer:

Greetings! Thank you for taking the time to review the manuscript (animals-2608904). Based on your valuable comments, I have revised the manuscript according to your comments, and the specific revisions are as follows:

  1. Please add more absolute numbers like the parameter combination values etc. to the abstract and conclusions sections to improve its information content rather than state vague trends.

Answer: Thank you for the valuable suggestions from the reviewers, as per your suggestions, the abstract and conclusion have been revised in the revised manuscript, as detailed in lines 37-61 and 851-871 of the revised manuscript.

  1. The introduction section could be extended to clarify the novelty of this manuscript further. More literatures are needed.

Answer: Thank you for the valuable suggestions from the reviewers. In accordance with their suggestions, the introductory part has been revised in the revised manuscript, as detailed in lines 65-113 of the revised manuscript.

Finally, I would like to thank the experts for their comments and suggestions that improved this manuscript and for you recognition of this article.

Looking forward to your reply. 

Kind regards,

Mr. Zhang,

E-Mail: [email protected]
